# The Global Deterioration Scale for Down Syndrome Population (GDS-DS): A Rating Scale to Assess the Progression of Alzheimer’s Disease

**DOI:** 10.3390/ijerph20065096

**Published:** 2023-03-14

**Authors:** Emili Rodríguez-Hidalgo, Javier García-Alba, Ramon Novell, Susanna Esteba-Castillo

**Affiliations:** 1Specialized Service in Mental Health and Intellectual Disability, Institute of Health Assistance (IAS), Parc Hospitalari Martí i Julià, 17190 Girona, Spain; 2Neurodevelopmental Group [Girona Biomedical Research Institute]-IDIBGI, Institute of Health Assistance (IAS), Parc Hospitalari Martí i Julià, 17190 Girona, Spain; 3Research and Psychology in Education Department, Complutense University of Madrid, 28040 Madrid, Spain

**Keywords:** global deterioration scale, rating scale, cognitive decline, cognitive testing, Alzheimer’s disease, dementia, down syndrome, intellectual disability

## Abstract

The aim of this study is to adapt and validate the global deterioration scale (GDS) for the systematic tracking of Alzheimer’s disease (AD) progression in a population with Down syndrome (DS). A retrospective dual-center cohort study was conducted with 83 participants with DS (46.65 ± 5.08 years) who formed the primary diagnosis (PD) group: cognitive stability (*n* = 48), mild cognitive impairment (*n* = 24), and Alzheimer’s disease (*n* = 11). The proposed scale for adults with DS (GDS-DS) comprises six stages, from cognitive and/or behavioral stability to advanced AD. Two neuropsychologists placed the participants of the PD group in each stage of the GDS-DS according to cognitive, behavioral and daily living skills data. Inter-rater reliability in staging with the GDS-DS was excellent (ICC = 0.86; CI: 0.80–0.93), and the agreement with the diagnosis categories of the PD group ranged from substantial to excellent with κ values of 0.82 (95% CI: 0.73–0.92) and 0.85 (95% CI: 0.72, 0.99). Performance with regard to the CAMCOG-DS total score and orientation subtest of the Barcelona test for intellectual disability showed a slight progressive decline across all the GDS-DS stages. The GDS-DS scale is a sensitive tool for staging the progression of AD in the DS population, with special relevance in daily clinical practice.

## 1. Introduction

Life expectancy in people with Down syndrome (DS) has increased considerably, to up to sixty years on average [1], evidencing an accelerated aging phenotype [2]. The risk of developing Alzheimer’s disease in people with DS (AD-DS) is greatly elevated due to the gene overexpression produced by the chromosome 21 trisomy, in which amyloid precursor protein and other gene modulations involve an accumulation of extracellular amyloid beta and neurofibrillary tangles [3,4]. AD neuropathology is present in more than 60% of people with DS by the age of 40 to 50 years [5]. Therefore, people with DS are considered at absolute risk of developing AD [6] and become a genetic paradigm for translational research with regard to the general population [7]. Despite these conceptual advances, AD-DS is usually diagnosed around the age of 50 years [8,9], later than expected [10], and these individuals are usually excluded from the clinical trials of typically developing individuals with AD [11].

Some major stages of the natural course of AD-DS have been suggested. Diagnostic categories in some studies are as follows: asymptomatic, cognitively stable, preclinical, prodromal AD, AD dementia, and uncertain [12,13,14]. Recently, under the biological definition and diagnosis relying on biomarkers, five stages have been proposed for AD-DS [15]: cognitively stable (CS), preclinical AD, mild cognitive impairment (MCI), prodromal AD and dementia. Bearing these data in mind, it seems self-evident that there is a lack of consensus on the proposed stages to capture the entire clinical continuum of AD-DS which, on the contrary, are based primarily on biological criteria. Furthermore, recent studies in the general population suggest different initial stages on the dementia continuum. In this sense, a subjective cognitive decline (SCD) stage has been proposed as a risk condition announcing possible dementia [16,17], but it must be supported by objective cognitive impairment to be considered part of the AD continuum [18]. Additionally, mild behavioral impairment (MBI) [19,20] has been proposed as a prodromal neuropsychiatric stage of emergent dementia. Due to the high risk of developing AD in subjects with DS, and considering that AD-DS pharmacological interventions are not efficient enough [21,22], it is necessary to arrange a grading scale that captures the whole clinical spectrum of the AD-DS with the inclusion of these recently proposed diagnostic categories to detect early therapeutic windows.

Although the biological perspective research framework has meant a forward leap in the prediction of MCI and AD, cognitive markers continue to be essential. There are several cases in daily clinical practice in which neuroimaging and biomarker findings suggest a certain diagnosis but without being supported by a neuropsychological examination, as in cases of biomarker positivity for AD without associated or progressive cognitive impairment [18]. Because biomarkers may fail in the diagnosis of AD, a neuropsychological examination can provide crucial cognitive information in the diagnostic and follow-up processes of the ID population [23]. In this sense, the aspects that should be explored in any longitudinal assessment for the entire continuum of AD-DS are dysexecutive cognitive or behavioral features, episodic memory, orientation, and general global cognitive decline [24,25,26,27]. 

The high prevalence of MCI and AD in individuals with DS has resulted in much research concerning the detection of both diagnoses. In a recent paper, the diagnostic criteria to delimit MCI in people with DS (MCI-DS) have been proposed [28], a stage that can be considered as prodromal of AD [29]. These authors showed that scores in the behavior rating inventory of the executive function parent form (BRIEF-P) [30] combined with scores on abstract thinking and verbal memory are useful values for detecting MCI-DS. Furthermore, the memory, language, and communication sections of the National Task Group-Early Detection Screen for Dementia (NTG-EDSD) seem to be sensitive to MCI-DS [31]. The Cambridge Cognitive Examination for Older Adults with Down’s syndrome—Spanish version (CAMCOG-DS) [32], with different cut-offs points, has been provided to detect prodromal AD or MCI and AD in people with DS and mild or moderate ID [13,32]. Furthermore, visuospatial-paired associate memory, hand–eye coordination, and semantic verbal fluency may be relatively sensitive events in the prodromal stage of AD-DS [10]. Finally, the performance of episodic memory using a modified cued recall test discriminates between individuals with preclinical, prodromal, and clinically manifest AD, albeit of mild degree [33]. Compared with the aforementioned major phases, the later stages of the AD-DS spectrum have not received special attention nor an international official definition. 

The progression of the disease in the clinical setting is assessed through cognitive, behavioral, psychiatric, and clinical tests. Additionally, rating scales provide a common language to diagnose, monitor, and evaluate therapeutic interventions. A good rating scale would be practical and properly validated in the target population, embracing different domains beyond cognition, and be sensitive to change in all stages to measure therapeutic effects [34], both pharmacological and non-pharmacological. In the general population, there are two reference scales for grading dementia. One of them is the clinical dementia rating scale (CDR) [35], a semi-structured interview with a patient and a close caregiver, covering six domains (memory, orientation, judgment and problem solving, work in the community, performance at home and in hobbies, and personal care). The results are distributed on a five-point scale from cognitive normality to severe dementia. Interestingly, an adaptation of the CDR, the CDR for frontotemporal lobar degeneration (CDR-FTDL) [36], was developed with the inclusion of language and behavior domains not contemplated in the original CDR, and has been shown to be sensitive in distinguishing disease progression between FTLD and AD in the general population [37]. Additionally, a modified CDR for adults with DS based on questionnaires and patient/caregiver interviews is available [38], capturing the progressive deterioration of AD in this population. 

The second grading dementia scale is the global deterioration scale (GDS [39], which determines the degree of functional loss based on the severity of the cognitive impairment. It consists of descriptions of the clinically differentiated stages of the AD continuum, from stage 1 (normal) to 7 (severe AD). Subsequently, the functional assessment staging (FAST) [40] provides the GDS with a division of phases 6 and 7, involving the progressive inability to maintain basic activities of daily living. The GDS is easier to apply than the CDR as it correlates with neuropathological, functional, global, and cognitive changes in the progression of MCI to AD [41] and with the hippocampal volume [42]. Furthermore, in our context, the GDS is the instrument of choice that determines the introduction and withdrawal of pharmacological and non-pharmacological treatment in patients with AD [43,44]. Therefore, it would be desirable to arrange a global deterioration rating scale which contemplates the amnesic [45,46] and the behavioral [47,48] forms of onset classically described in AD-DS. Consequently, such a scale would improve both the response to health needs according to the time point of disease progression and provide researchers in this field with a common language in our context.

The main objective of the present study is to examine the feasibility of an adapted GDS for its use in people with DS on the continuum of AD. We expect to find that the stages proposed for GDS-DS are anchored to the performance of cognitive and behavioral instruments that are well established in people with DS and a mild or moderate level of ID. 

## 2. Materials and Methods

### 2.1. Study Design and Description of the Sample

This is a retrospective, dual-center cohort study of Caucasian adults with DS. A total of 87 participants were identified from the Servicio Especializado en Salud Mental y Discapacidad Intelectual (Specialized Mental Health ID Unit, Institute of Health Assistance, Girona) and the Unidad de Adultos con Síndrome de Down (Adult Down Syndrome Unit, La Princesa University Hospital, Madrid, Spain). 

A neurological, psychiatric, and laboratory examination was applied to all the participants. The neurological examination consisted of taking the participant’s history (e.g., previous central nervous system alterations, relevant drug treatment, substance abuse, sleep disorders) and a physical examination. Psychiatric data were collected using the psychiatric assessment schedule for adults with a developmental disability (PAS-ADD) [49]. Blood samples were obtained in order to detect hypothyroidism, vitamin B12 deficiency, and anemia. 

The inclusion and exclusion criteria of the study are displayed in Table 1. Going into detail, the age cut-off was chosen based on previous studies which reported that being above 39 years of age represents a high risk of developing cognitive decline from a previous level of efficiency in people with DS [50]. The level of ID was determined according to the DSM-5 criteria [51]. All participants were required to have a reliable informant available to report on the present and past adaptive skills and behavior of the participants. 

Those with sensory impairments that prevented the completion of the research protocol, those with a history of alterations of the central nervous system (e.g., brain tumors, head injury, stroke), those with uncontrolled sleep disorders (e.g., obstructive sleep apnoea), and individuals suffering from substance abuse were excluded. Patients were also excluded if they had untreated anemia, vitamin B12 deficiency, or uncontrolled hypothyroidism because of their potential risks of influencing behavior. 

It should be noted that cases of conduct disorder, depression, and anxiety were not automatically ruled out. Those cases caused by a stressful event in the past six months, were excluded but those (un)treated cases related to normal daily functioning were not, according to clinical judgment. 

Finally, participants exposed to a high anticholinergic burden through polypharmacy from psychotropics, gastrointestinal and cardiovascular medications were excluded if the treatment was considered ineffective, according to clinical judgment. 

Our study was conducted in accordance with The Code of Ethics of the World Medical Association (Declaration of Helsinki). The study protocol was approved by the Clinical Research Ethical Committee of the Parc Hospitalari Santa Caterina (Girona). Written informed consent was obtained from parents and written and pictorial assent were additionally obtained.

### 2.2. Clinical Assessment

The protocol consisted of a cognitive test and behavioral and daily living skills questionnaires that have been shown to be most sensitive to cognitive impairment in Spanish-speaking subjects with DS. 

#### 2.2.1. Cognitive Assessment

The protocol with the following test was applied to all the participants: The level of ID was based on the results from the Kaufman brief intelligence test, second edition (KBIT-II) [52] and Vineland adaptive behavior scales–second Edition (Vineland II) [53].The *KBIT-II* is a test designed for the measurement of verbal and non-verbal intelligence. It consists of two sub-tests that assess crystallized intelligence and fluid intelligence and allow the establishment of the level of ID.The *Vineland II* scale measures adaptive behaviors, including communication, daily living skills, socialization, and motor skills.General cognitive abilities were assessed using the following scales: the Cambridge cognitive examination for older adults with Down’s syndrome and other intellectual disabilities—Spanish version (CAMCOG-DS) [32], and the Barcelona test for intellectual disability (BT-ID) [54].The CAMCOG-DS comprises seven cognitive domains (orientation, language, memory, attention, praxis, abstract thinking, and perception). The maximum score is 109 points. The psychometrical properties are good: test–retest reliability = 0.92, ICC = 0.91, internal consistency = 0.70 − 0.93, and κ values of 0.95 and 0.97 versus DSM-IV [51] and ICD-10 [55] criteria, respectively. It is a reliable tool for the assessment of cognitive impairment in people with ID with and without DS with mild and moderate levels of ID.The *BT-ID* consists of 67 subtests grouped into eight cognitive domains (language, working memory, orientation, praxis, attention, executive function, visuoconstruction, and memory). It shows good psychometric properties: test–retest reliability = 0.91, ICC = 0.95, internal consistency = 0.70 − 0.93. It provides normative data for five groups based on intellectual disability level, age, and curricular competence.Planning and problem solving were assessed with the Tower of London—Drexel University: 2nd edition ID version [56]. Its psychometric properties are reliable for differentiation between subjects with mild and moderate ID, and are associated with other measures of executive functions. Additionally, it demonstrated sufficient evidence of reliability and validity in adults with Down syndrome.

#### 2.2.2. Informants’ Questionnaires

The behavior rating inventory of executive function, parents’ form (*BRIEF-P*) [30] measures executive functions or self-regulation in their everyday environments. It consists of two indexes: (1) Behavioral regulation index (BRI), composed of inhibit, shift, and emotional control scales; and (2) the metacognitive index (MI), composed of initiate, working memory, planning, organization, and monitor scales.The informant interview of the Cambridge examination for older adults with Down’s syndrome and other intellectual disabilities—Spanish version (CAMDEX-DS) [32]. It consists of (1) a structured interview with the informant/family member to collect information to detect a decline in the individual’s best level of functioning, their cognitive and functional impairment, and the mental and physical health of the participant, including depression, anxiety, paranoid symptoms, delirium, substance abuse, physical disability, hypothyroidism, cerebrovascular problems, and pharmacological treatment; (2) the CAMCOG-DS (see the cognitive test scale section in this article); (3) a guide for the clinical diagnosis of AD, capturing changes in daily adaptive skills (section A), memory (section B), other cognitive domains (section C1), personality/behavior (section C2) and confusional acute syndrome (section D); and (4) suggestions for the correct intervention in people with ID and dementia. In this study, the CAMDEX-DS was used only for clinical diagnostic purposes to avoid the risk of circularity.

The participants of the PD group were classified according to three diagnostic categories: cognitive stability, MCI and AD. The diagnosis of cognitive stability was made when participants had no cognitive impairment or decline in adaptive skills, according to the CAMDEX-DS informant section. As there is currently no internationally accepted official definition for MCI-DS [57], the diagnosis of MCI or AD was based on expert multidisciplinary clinical judgment according to recent publications, as is recommended in standard practice for DS [26,28,58,59,60,61]. In detail, a participant fulfilled the diagnosis of MCI when presented with a single or multiple cognitive decline and/or loss of functionality according to the CAMDEX-DS informant section [26]. The diagnosis of AD was obtained if participants had memory decline or another cognitive impairment, such as aphasia, apraxia, agnosia, or dysexecutive syndrome, or loss of functionality [62]. In both conditions, the information about changes from previous levels of performance must be supported by a close caregiver [26,58,60]. It should be noted that the CAMDEX-DS was used only for diagnostic purposes, whereas performance on the neuropsychological tests confirmed the diagnosis through a longitudinal clinical study [26]. 

#### 2.2.3. Global Deterioration Scale for Down Syndrome (GDS-DS)

The GDS-DS began at stage 1 (cognitive and behavioral stability), in which subjects were placed if they had neither subjective complaints nor cognitive or behavioral impairments identified by the instruments of the study protocol. Stage 2 (subjective cognitive and/or behavioral impairment) included those subjects with changes in cognition, behavior or adaptive skills, as self-reported or reported by caregivers but not supported by objective data. Stage 3 (mild cognitive and/or behavioral impairment) corresponded to those cases with reports of cognitive or behavioral impairment by the patient or confirmed by a reliable informant, supported by objective data, with no or very mild loss of adaptive skills. Stages 4 (mild Alzheimer’s disease), 5 (moderate Alzheimer’s disease) and 6 (advanced Alzheimer’s disease) were reserved for those subjects presenting mild to severe deterioration in cognitive and behavioral domains, as reported by an informant, supported by objective data and demonstrating affected adaptive skills (Table 2). 

GDS-DS adaptation

The GDS-DS was constructed on the basis of the global deterioration scale (GDS) [39]. Some modifications were introduced to adapt this scale to the study population. First, the original GDS scale consists of seven stages (1–7), whereas the GDS-DS consists of six stages (1–6). Secondly, because growing evidence suggests that subjective cognitive decline [16,17] and mild behavioral impairment [19,20] could predict AD, subjective cognitive and behavioral aspects were considered. Therefore, stage 2 was labeled as subjective cognitive and/or behavioral impairment and stage 3 as mild cognitive and/or behavioral impairment.

In general, for the delimitation of the GDS-DS stages, the performance of the participants in the cognitive instruments and the information of the behavioral questionnaire and caregivers gathered from the study protocol were classified as mandatory or supporting criteria, according to the recent literature about AD and DS (Table 2).

Mandatory criteria

The mandatory criteria were established based on those cognitive and behavioral instruments that had obtained sufficient quantitative data to differentiate MCI and AD, in addition to adaptive skills. The purpose was to establish criteria that were as inclusive as possible; thus, the following instruments and scores were considered:Behavioral Regulation Index—BRIEF-P [30]. This questionnaire demonstrates that, in people with DS, a cut-off point of ≥ 55 allows the classification of stable subjects at the cognitive and behavioral levels (sensitivity = 90), while a cut-off point of <32 allows the classification of subjects with MCI (specificity = 0.90). For scores between 32 and 55, the diagnosis is more doubtful [28]. According to these data, the following cut-off points were assigned for our study: ≥33 would be a criterion for cognitive and behavioral stability (stage 1) and subjective cognitive and/or behavioral impairment (stage 2), and ≤32 would be a criterion for mild cognitive or behavioral impairment (stage 3) and mild, moderate and advanced AD (stages 4, 5 and 6, respectively).Regarding the total score of the *CAMCOG-DS* [32], it has become clear that a cut-off point of the total score = 68 allows the detection of AD in subjects with DS and mild ID (sensitivity: 80%; specificity: 81%), and a cut-off point of 52 (sensitivity: 85%; specificity: 81%) in those with moderate ID [32]. Additionally, in another study of subjects with DS and mild ID, a cut-off point of 82 (sensitivity: 80%, specificity 80.5%) differentiated between asymptomatic and prodromal AD, and a cut-off point of 80 (sensitivity: 75%, specificity 87.8%) differentiated asymptomatic and dementia AD. In the same study, for those with moderate ID, a cut-off point of 64 (sensitivity: 66.7%, specificity 72.3%) differentiated between asymptomatic and prodromal AD, and a cut-off point of 56 (sensitivity: 84%, specificity 84.3%) between asymptomatic and dementia AD [13]. The data of these two previous studies were incorporated for all the stages of the GDS-DS, according to the ID level and the CAMCOG-DS total score.The Guide to clinical diagnosis of the CAMDEX-DS—Spanish version [32]. The decline of daily life skills is decisive in the diagnostic process of AD, since in the MCI, these are preserved or only very mildly affected. Therefore, in our study, positive scores in sections B (memory), section C1 (other cognitive domains) or section C2 (personality/behavioral) were considered indicative of mild cognitive and/or behavioral impairment (stage 3). Furthermore, positive scores in section A (daily living skills), section B and sections C1 or C2 and negative scores in section D (acute confusional syndrome) are indicative of AD (stages 4, 5 and 6).

It should be noted that the BRIEF-P (BRI index) and the CAMCOG-DS, in addition to providing a quantitative basis for each stage of the GDS-DS, also cover the symptoms of the two most frequent forms of AD onset in the DS population, either the amnestic [45,46] or the behavioral variants [47,48]. 

Supporting criteria

The supporting criteria were established based on those cognitive and behavioral instruments of the protocol study that demonstrated a decrease associated with an increase in the clinical intensity of AD-DS from the MCI-DS phase. These criteria might not be mandatory, but their presence helped determine the stage of the GDS-DS that would be assigned to each participant. The instruments considered for this purpose were: BT-ID: the orientation, semantic fluency (eating and drinking), formal fluency, delay verbal memory (stories) and visual discrimination subtests.CAMCOG-DS: abstract thinking subtest.

Recent studies claim that significantly lower scores on the above subtests can be indicative of MCI-DS and the early phase of AD [10,13,26,28]. Therefore, these were added as supporting criteria for AD-DS from stage 3 (mild cognitive and/or behavioral impairment) upward (stages 4, 5 and 6). 

It should be emphasized that in order to identify the stages of AD (4, 5 and 6), in addition to the mandatory and supporting criteria established, the degree of deterioration (mild, 4; moderate, 5; and advanced, 6) would have to be determined by the judgment and clinical experience of specialists, who play a decisive role in this regard. 

For stage 6 (advanced Alzheimer’s disease), an additional criterion was established. Based on our clinical experience and other studies of populations with DS, it has been suggested that some tests cannot be administered to those with advanced stages of AD or in populations with severe intellectual disabilities [57] because they do not provide enough information due to the “floor effect” [22]. Therefore, the criterion of not being able to administer all or any cognitive tests from the screening protocol to participants would, on its own, be an indicator of advanced AD, supported by affected adaptive skills.

### 2.3. Procedures

Cognitive, behavioral and adaptive skills data from the PD group were used retrospectively to place each participant in a stage of the GDS-DS and thus form the GDS-DS group. Two neuropsychologist specialists in DS blinded to the diagnosis of the PD group (SEC, JGA) placed each subject into one of the six levels of the proposed GDS-DS rating scale, according to the level of ID of each participant and the mandatory and supporting criteria established (Table 1). It should be highlighted that in order to avoid circularity, the two specialists were different to those who made the diagnoses for the PD group. The degree of agreement between the examiners in placing the subjects in the stages of the GDS-DS was checked. The inter-rater reliability test was applied to the two raters’ first GDS-DS classifications. Then, the classifications of the two raters was transformed into single values as follows: if they matched the GDS-DS assignment, the same value was maintained, and when the two raters did not agree on the classification using the GDS-DS, the case was discussed until a consensus was reached. Additionally, the demographic and the diagnostic categories of the PD and the GDS-DS groups were analyzed in order to observe possible similarities. 

The agreement between the two raters classifying the participants using the GDS-DS compared with the PD group was analyzed. For this purpose, the stages 1 (cognitive and behavioral stability) and 2 (subjective cognitive and/or behavioral impairment) of the GDS-DS were associated with the cognitive stability diagnosis of the PD group; stage 3 (mild cognitive and/or behavioral impairment) of the GDS-DS was associated with the diagnosis of mild cognitive impairment of the PD group; and stages 4, 5 and 6 (Mild, Moderate and Advanced Alzheimer’s disease) of the GDS-DS were associated with the Alzheimer’s disease of the PD group. Codes associated with each category of the PD group are displayed in Table 3. 

Finally, as the progressive deterioration of adaptive skills was an immovable criterion and an analysis with this variable could lead to a circularity problem, the authors checked for a possible association of the GDS-DS stages with a selection of the cognitive and behavioral instruments included as mandatory and supporting criteria for each stage. 

### 2.4. Statistical Analysis

We performed all statistical analyses using the software program SPSS (version 27.0; SPSS Inc., Chicago, IL, USA). Bilateral significance levels were set at a *p*-value of less than 0.05. The normality of data was assessed using the Kolmogorov–Smirnov test, and subsequently, non-parametric analyses were carried out. For the GDS-DS group, demographic characteristics, level of ID, and performance on selected neuropsychological tests were analyzed by a non-parametric Kruskal–Wallis test followed by the Bonferroni correction for post hoc pairwise comparisons or Pearson’s chi-square test for category data. Spearman’s correlation analysis was used to determine the associations between the stages of the GDS-DS with the cognitive and behavioral data selected. A Mann–Whitney U test was performed with pairs of the GDS-DS groups because the effect size between the cognitive and behavioral performance across the GDS-DS stages was computed using the non-parametric probability of superiority estimation (PSest), interpreted as small (≥56), medium (≥0.64), and large (≥0.71) [63]. Inter-rater agreement and concordance between the stages issued by rater 1 and rater 2 with the primary diagnosis of the study were analyzed by using Cohen’s weighted kappa values. 

## 3. Results

### 3.1. Sample and Demographics

Of the subjects selected, four ultimately did not agree to participate in the study. The final sample formed the primary diagnosis (PD) group and consisted of 83 subjects (46.65 ± 5.08 years; male = 46 (55.4%), female = 37 (44.6%)) and was divided into three groups: (1) cognitively stable group, with 48 subjects (45.10 ± 3.83 years; male = 24 (50%), female = 24 (50%)); (2) mild cognitive impairment group, with 24 subjects (47.46 ± 5.35 years; male = 16 (66.7%), female = 8 (33.3%)); and (3) Alzheimer’s disease group, with 11 subjects (51.64 ± 6.05 years; male = 6 (54.5%), female = 5 (45.6%)). Seven of the initial candidates had to be excluded because they could not undergo the neuropsychological assessment.

The demographic details of the PD group are reported in Table 4. Of the total 83 participants, about half were diagnosed as being cognitively stable, one third as having mild cognitive impairment and a smaller number as having Alzheimer’s disease. The mean age of the subjects with AD was significantly higher than the stable subjects but not than the MCI group. No statistical differences were observed in the gender and level of ID across the groups. 

Within the GDS-DS group (Table 5), 27 participants were diagnosed as having cognitive stability (stage 1) and with subjective cognitive and/or behavioral impairment (stage 2), 25 with mild cognitive and/or behavioral impairment (stage 3) and 31 participants with any degree of AD (stages 4, 5 or 6). Interestingly, comparing these two groups, a similar number of participants were classified as MCI with both methods and 20 more subjects as AD. There were more participants diagnosed as more cognitively stable in the PD group than those placed in stage 1 of GDS-DS (48 and 9, respectively). As expected, subjects diagnosed with AD in the PD group and those placed on the stage 6 of the GDS-DS (advanced Alzheimer’s disease) were the oldest. 

### 3.2. Cognitive and Behavioral Data across the GDS-DS Stages

These analyses were conducted without the data from the stage 6 subjects, who were excluded because they could not complete the study protocol.

The analysis of the correlations revealed negative significant correlations between performance on the tests and the GDS-DS, except for the BRIEF-P and the visual discrimination BT-ID subtest. The correlations were moderate between the GDS-DS and the CAMCOG-DS Total score, orientation subtest and formal fluency, but weak with abstract thinking subtest of the CAMCOG-DS, free memory delay (stories) and semantic fluency of the BT-ID (Table 6).

The performance on the cognitive and behavioral tools across the GDS-DS stages is displayed in Table 7. Overall, the performance on the CAMCOG-DS total score and the orientation subtest (BT-ID) decreased significantly across stage 1 (cognitive and/or behavioral stability) to 5 (moderate Alzheimer’s disease). 

In terms of sectors, performance significantly decreased regarding: Delay stories (BT-ID) between stage 2 (subjective cognitive and/or behavioral impairment) and 5 (moderate Alzheimer’s disease) and stages 3 (mild cognitive and/or behavioral impairment) and 5 (moderate Alzheimer’s disease).Formal fluency (TB-ID) between stage 1 (cognitive and/or behavioral stability) and 5 (moderate Alzheimer’s disease), stage 2 (subjective cognitive and/or behavioral impairment), stage 3 (mild Alzheimer’s disease) and stage 4 (moderate Alzheimer’s disease).Semantic fluency (BT-ID) between stages 2 (subjective cognitive and/or behavioral impairment) and 5 (moderate Alzheimer’s disease).

Note that (i) the performance on abstract thinking, orientation, delay stories, and formal fluency subtest (BT-ID) did not decrease between stage 1 (cognitive and behavioral stability) and 2 (subjective cognitive and/or behavioral impairment); (ii) the performance on the BRI index (BRIEF-P) decreased between 1 (cognitive and behavioral stability) and 3 (mild cognitive impairment), but was heterogeneous from stage 3 (mild cognitive and/or behavioral impairment) to stage 5 (moderate Alzheimer’s disease); (iii) the performance on delay stories subtest (BT-ID) did not show significant oscillations between stage 2 (subjective cognitive and/or behavioral impairment) and 3 (mild cognitive and/or behavioral impairment); and the performance on the visual discrimination subtest (BT-ID) was similar across all the stages. These data had an impact on the posterior analysis of the effect sizes values. 

The raw differences that showed a progressive decrease between performances on the selected tests across the GDS-DS stages are displayed in Table 8 and Appendix A of the online Appendix A.

Overall, small to large effect sizes were observed between stage 2 (subjective cognitive and/or behavioral impairment) and 5 (moderate Alzheimer’s disease) for the CAMCOG-DS total score and orientation subtest (BT-ID). 

The effect sizes ranged from small to medium between stages 1 (cognitive and behavioral stability) and 3 (mild cognitive and/or behavioral impairment) for the BRI index (BRIEF-P). Medium to large effect sizes were observed for the formal fluency subtest (BT-ID) between stage 2 (subjective cognitive and/or behavioral impairment) and stage 4 (mild Alzheimer’s disease). 

The effect sizes ranged from small to medium for free delay memory and semantic fluency subtest (BT-ID) between stage 4 (mild Alzheimer’s disease) and 5 (moderate Alzheimer’s disease). 

A small effect size was observed between stage 3 (mild cognitive and/or behavioral impairment) and 4 (mild Alzheimer’s disease) on the *visual discrimination* subtest (BT-ID).

### 3.3. Reliability

The inter-rater reliability for staging using GDS-DS was excellent, with a mean Cohen weighted κ value of 0.86 (CI: 0.80–0.93). The agreement between the two raters classifying the PD group using the GDS-DS was excellent, as determined by specialists, with κ values of 0.82 (CI: 0.73–0.92) and 0.85 (CI: 0.77–0.94).

## 4. Discussion

The main goal of the present study was to devise a global rating scale for AD in people with DS based on the GDS scale [39]. The resulting GDS-DS amplifies the classical phases of the AD-DS (stability, prodromal AD and AD). Our purpose was to provide a scale that captures progressive decline in the entire AD-DS continuum. The GDS-DS meets this requirement from the cognitive stability to AD stages, especially in relation to performance on the CAMCOG-DS *total score* and *orientation* subtests of the BT-ID and dementia.

The GDS-DS ranges through six stages from cognitive and behavioral stability (stage 1) to advanced Alzheimer’s disease (stage 6), contrasting with the seven stages of the global deterioration scale (GDS) [39] for the general population. In view of the results obtained in the present work, in which an overall decrease in deterioration is detected by the CAMCOG-DS total score, six stages capture the entire continuum of AD in people with DS and, in addition, they are consistent with the stages of AD proposed in research of the National Institute on Aging and the Alzheimer’s Association (NIA-AA) [6]. 

The six stages of the GDS-DS differ from the five stages of the clinical dementia rating scale (CDR) [35] and the modified clinical dementia rating scale for people with Down syndrome questionnaire (CDR-QDS) and interview (CDR-IDS) forms [38]. These two rating scales (CDR and CDR-IDS/CQR-IDS), as opposed to our GDS-DS, do not include stage 2 (subjective cognitive and/or behavioral impairment). This can lead to misdiagnosis because subjective cognitive impairment is increasingly seen as an early symptom of dementia risk in the general population [16,17] and although it is not a sensitive aspect by itself [18], it must be considered with complementary clinical data as a different entity from MCI. In addition, behavioral aspects are also not considered in these scales, although they are important factors for detecting possible changes associated with prodromal AD in subjects with DS [47]. Furthermore, in our clinical dementia contexts, the global deterioration scale (GDS) [39] is used to monitor, design and modify appropriate pharmacological treatment in patients with AD in the general population [43,44]. Thus, the authors consider the GDS-DS a closer instrument for daily clinical practice and recommend its use for clinical trials in subjects with AD-DS. 

Seven participants were placed in the stage 6 (advanced Alzheimer’s disease) category of the GDS-DS in our study. The criterion to be placed in this stage was that the subject did not endure the neuropsychological examination proposed in the protocol and this was, in turn, an exclusion criterion for the PD group on which the present work is based. Some DS adults in the symptomatic stages of AD are not able to complete a neuropsychological examination [22]. The authors of the present study agree, as routine follow-up visits have encountered this setback. As a result, despite progress in terms of instruments adapted for people with ID, the neuropsychological tests used in the aforementioned studies and in daily clinical practice are not sufficiently valid to capture the deterioration in advanced stages of AD-DS nor for severe/profound intellectual disability [57,64,65]. Therefore, in terms of daily clinical practice, for people placed in stage 6 in our study, the authors recommend the use of adapted tools, such as the cognitive exploration scale for people with intellectual disability and extended support (ECDI-SE) or the modified ordinal scales of psychological development for people with intellectual disability (M-OSPD-ID). However, at the same time, this also means that using these instruments in the early stages of AD can lead to a ceiling effect on the scores. Future studies should focus on valid instruments for all phases of AD, avoiding ceiling and floor effects, regardless of the severity of symptoms.

Performance in the orientation subtest of BT-ID decreased significantly throughout the GDS-DS stages. Although not pathognomonic of AD, temporal disorientation is frequently observed in daily clinical practice in subjects with cognitive impairment. This requires semantic and episodic information activation [66] and has been linked to atrophy in the posterior hippocampus [67] and the disconnection between the posterior part of the right medial temporal gyrus and the posterior cingulate cortex [68]. The progressive loss of orientation has already been described in the transition from MCI to AD in people with DS [26]. In view of these facts, this subtest should be present in longitudinal follow-ups in individuals with ID.

Performance on executive functions (abstract thinking, free delay memory (stories), semantic and formal fluency) shows a slight decline with modest sustained effect sizes in some stages of the AD continuum. The anatomical substrate of these functions is linked with the temporoparietal, precuneus-posterior cingulate and occipital areas [24], which show a decreased volume [41,60] and loss of integrity in white matter tracts [58,69] in individuals with AD-DS. In addition, an increase in alpha band synchronization using magnetoencephalography has been found in the functional connectivity in the AD continuum in the general and in the DS population and is associated with cognitive decline in executive function, language and working memory [62]. Additionally, MCI-DS individuals with confirmed amyloid positivity who progress to AD have shown a pattern of increased delta activity in frontal regions [70]. The clinical relevance of these findings suggests that all these cognitive functions also should be examined in longitudinal clinical follow-ups 

Declines in the CAMCOG-DS total score have been shown to be related to changes in all of the stages of the GDS-DS. A decline in performance on CAMCOG-DS has been found in the entire AD-DS continuum [13,26,28,62], regardless of whether the level of ID is mild or moderate [22]. Interestingly, performance on CAMCOG-DS is linked to amyloid deposition [71], and it is recommended that longitudinal studies assess cognitive changes related to ID and dementia [72]. Therefore, CAMCOG-DS could be a suitable instrument for anchoring the GDS-DS stages of our context, in a similar way that the mini examen cognoscitivo (MEC) [73] is anchored to the global deterioration scale [39] in the general population. 

The *BRI* index of BRIEF-P [30] does not change significantly across the GDS-DS stages, but it drops slightly from the cognitive/behavioral stability to mild cognitive and/or behavioral impairment, stages 1 to 3, respectively. The authors selected the *BRI* index because worse scores on this index could differentiate between healthy and MCI-DS subjects [28], and our results partially replicate this. In light of these results, the inclusion of behavioral impairment in the GDS-DS seems appropriate. In addition, in the general population, mild behavioral impairment [19] improves the specificity of MCI as an at-risk state for incident dementia [74] and has been associated with higher AD polygenic risk scores [75]. Considering that behavioral changes are able to herald AD in the DS population [47], the authors recommend the use of the *BRI* index scores to detect behavioral impairment in the DS population in the earlier stages. 

The main strength of the present study is that the GDS-DS, with the inclusion of behavioral aspects, allows the capture of subjects in the initial stages of AD-DS either due to memory or behavioral difficulties, in addition to being associated with performance in cognitive and behavioral tests that have been shown to be reliable in the DS population.

Additionally, the GDS-DS could be the first step in unifying the criteria for classifying the continuum of Alzheimer’s disease in people with DS. It allows a common language for communication between clinicians and researchers and could be a basic instrument for the selection of samples in clinical trials in people with DS.

Furthermore, the GDS-DS, can also be used to inform families about the stages of Alzheimer’s disease, when sufficient data from future studies have been collected to construct specific clinical profiles in combination with functional assessment staging. Knowing the continuum can complement the neurological diagnosis and facilitate family members’ understanding of the expected prognosis.

However, some limitations have to be acknowledged. In this study, we used cross-sectional information that relied on the longitudinal data of a three-year follow-up study. We first wanted to verify that the GDS-DS was applicable to people with DS. As this has been proven to be feasible, future studies could investigate the application of the scale in longitudinal follow-ups to minimize possible cohort effects.

Additionally, our study was based on data collected from neuropsychological tests, behavioral questionnaires and informant interviews. Today, neuroimaging or neurophysiological techniques provide rich complementary data that correlate brain changes with cognitive features in aging and AD. Thus, future studies must include the linking of neuroimaging and neurophysiological data with the stages of GDS-DS. 

## 5. Conclusions

In summary, GDS-DS is not designed as a diagnostic tool but as a quantitative measure of disability. The insights delivered in this paper show that the proposed GDS-DS rating scale represents an important attempt at staging people with DS throughout the continuum of AD, providing a unique method of classification that can be useful in clinical trials and in daily clinical practice. As noted in the development of the original GDS [39], the limits of each of the GDS-DS stages are not axiomatic, but they do allow graduation as a guideline that facilitates the monitoring of the AD-DS continuum. 

## Figures and Tables

**Table 1 ijerph-20-05096-t001:** Inclusion and exclusion criteria.

Inclusion Criteria	Exclusion Criteria
≥39 years old	Severe sensory impairments
Both sexes	No reliable informant
Mild or moderate intellectual disability	Untreated anemia
DS confirmed karyotype	Vitamin B12 deficiency
	Uncontrolled hypothyroidism
	Behavior disorder (comorbid, affecting normal functioning)
	Uncontrolled sleep disorders
	Substance abuse
	Previous central nervous system alterations
	Relevant drug treatment

**Table 2 ijerph-20-05096-t002:** Stages and diagnostic criteria for each stage of the GDS-DS.

Stages	
1	Cognitive and behavioral stability All must be present I. BRIEF-P BRI index: ≥33 II. CAMCOG-DS total: ≥83 (mild ID); ≥65 (moderate ID) III. Any CAMDEX-DS diagnostic criteria (no A, B, C1 or C2)
2	Subjective cognitive and/or behavioral impairment All must be present I. BRIEF-P BRI index: ≤33 II. CAMCOG-DS total: ≥ 83 (mild ID); ≥65 (moderate ID) III. CAMDEX-DS diagnostic criteria: B or C1 or C2
3	Mild cognitive and/or behavioral impairment (I and/or II) + III must be present I. BRIEF-P BRI index: ≤32 II. CAMCOG-DS total: ≤82 − 69 (mild ID), ≤64 − 57 (moderate ID) III. CAMDEX-DS diagnostic criteria: B or C1 or C2 ≤2 supporting criteria
4	Mild Alzheimer’s disease (I and/or II) + III must be present I. BRIEF-P BRI index: ≤32 II. CAMCOG-DS total: ≤68 (mild ID), ≤56 (moderate ID) III. CAMDEX-DS diagnostic criteria: A, B, C1 or C2, no D ≥2 supporting criteria
5	Moderate Alzheimer’s disease (I and/or II) + III must be present I. BRIEF-P BRI index: ≤32 II. CAMCOG-DS total: ≤68 (mild ID), ≤56 (moderate ID) III. CAMDEX-DS diagnostic criteria: A, B, C or C2, no D ≥3 supporting criteria
6	Advanced Alzheimer’s disease (I and/or II) + III must be present, or III and V I. BRIEF-P BRI index: ≤32 II. CAMCOG-DS total: ≤68 (mild ID), ≤56 (moderate ID) III. CAMDEX-DS diagnostic criteria: A, B, C1 or C2, no D ≥3 supporting criteria Incomplete or no administered neuropsychological examination

GDS-DS, global deterioration scale for people with Down’s syndrome; BRIEF-P BRI, behavior rating inventory of executive function, parents’ form behavioral regulation index; CAMCOG-DS, Cambridge cognitive examination for older adults with Down’s syndrome and other intellectual disabilities; ID, intellectual disability; CAMDEX-DS, Cambridge examination for older adults with Down’s syndrome and other intellectual disabilities.

**Table 3 ijerph-20-05096-t003:** Recode numeric values.

GDS-DS Stages	Primary Diagnosis
1. Cognitive and behavioral stability 2. Subjective cognitive and/or behavioral impairment	0. Cognitive stability
3. Mild cognitive and/or behavioral impairment	1. Mild cognitive impairment
4. Mild Alzheimer’s disease 5. Moderate Alzheimer’s disease 6. Advanced Alzheimer’s disease	2. Alzheimer’s disease

GDS-DS, global deterioration scale for people with Down’s syndrome.

**Table 4 ijerph-20-05096-t004:** Demographics of the primary diagnosis sample.

	Total	Cognitive Stability	Mild Cognitive Impairment	Alzheimer’s Disease	*p*
*n*	83	48	24	11	
Age	46.65 (39–63)	45.10 (39–43)	47.46 (39–61)	51.64 (43–63)	0.01 * a
Gender					0.41
Male	46 (55.4%)	24 (50%)	16 (66.7%)	6 (54.5%)	
Female	37 (44.6%)	24 (50%)	8 (33.3%)	5 (45.6%)	
ID level					0.61
Mild	49 (59.1%)	29 (60.4%)	15 (62.5%)	5 (45.6%)	
Moderate	34 (40.9%)	19 (39.6%)	9 (37.5%)	6 (54.6%)	

Age values are shown as means and range. For the sex and ID level, percentages regarding the group are shown. ID, intellectual disability. * *p* < 0.05, according to Kruskal–Wallis test with Bonferroni correction for age, or χ^2^-test for gender and ID level. a, between cognitive stability and Alzheimer’s disease.

**Table 5 ijerph-20-05096-t005:** Demographics for each stage of the global deterioration scale for people with Down’s syndrome.

		GDS-DS Stages						*p*
	Total	1	2	3	4	5	6	
		Cognitive/Behavioral Stability	Subjective Cognitive/Behavioral Impairment	Mild Cognitive/Behavioral Impairment	Mild Alzheimer’s Disease	Moderate Alzheimer’s Disease	Advanced Alzheimer’s Disease	
*n*	83	9	18	25	10	14	7	
Age	46.65 (39–63)	46.44 (40–61)	44.72 (39–54)	45.60 (39–54)	46.60 (42–52)	47.43 (41–54)	54.14 (49–63)	0.02 * a, * b
Gender								0.15
Male	46 (55.4%)	8 (88.9%)	11(61.1%)	10(40.0%)	4(40.0%)	9 (64.3%)	4 (57.1%)	
Female	37 (44.6%)	1 (11.1%)	7 (38.9%)	15 (60.0%)	6 (60.0%)	5 (35.7%)	3 (42.9%)	
ID level								0.08
Mild	49 (59.1%)	4 (44.4%)	15 (83.3%)	17(68.0%)	4 (40.0%)	6 (42.9%)	3 (42.9%)	
Moderate	34 (40.9%)	5 (55.6%)	3 (16.7%)	8 (32.0%)	6 (60.0%)	8(57.1%)	4 (57.1%)	

Age values are shown as means and range. For the sex and ID level, percentages regarding the group are shown. ID, intellectual disability. * *p* < 0.05, according to Kruskal–Wallis test with Bonferroni correction for age, or χ^2^-test for gender and intellectual disability level. a, between subjective cognitive/behavior impairment and advanced Alzheimer’s disease; b, between mild cognitive/behavior impairment and advanced Alzheimer’s disease.

**Table 6 ijerph-20-05096-t006:** Correlations between GDS-DS stages and cognitive/behavioral assessment.

Cognitive and Behavioral Assessment	Rho
BRIEF-P	
BRI index	−0.17
CAMCOG-DS	
Total score	−0.70 **
Abstract thinking	−0.39 **
BT-ID	
Orientation	−0.78 **
Free delay memory (stories)	−0.37 **
Semantic fluency (eat/drink)	−0.38 **
Formal fluency	−0.54 **
Visual discrimination	−0.11

GDS-DS, global deterioration scale for people with Down’s syndrome; BRIEF-P, behavior rating inventory of executive function parents’ form; BRI, behavioral regulation index; CAMCOG-DS, Cambridge cognitive examination for older adults with Down’s syndrome and other intellectual disabilities; BT-ID, Barcelona test for intellectual disability. ** *p* < 0.01, (two-tailed test), using Spearman’s correlation coefficient.

**Table 7 ijerph-20-05096-t007:** Cognitive and behavioral performance across the GDS-DS stages.

		GDS-DS Stages					*p*
		1	2	3	4	5	
	Total	Cognitive/Behavioral Stability	Subjective Cognitive/Behavioral Impairment	Mild Cognitive/Behavioral Impairment	Mild Alzheimer’s Disease	Moderate Alzheimer’s Disease	
BRIEF-P							
BRI index	40.99 (35–47)	47.00 (39–49)	40.00 (36–47)	36.00 (30–42)	40.00 (36–45)	37.00 (36–51)	
CAMCOG-DS							
Total score	72.00 (55–88)	**84.00 (76–87)**	**82.50 (76–87)**	**71.00 (60–78)**	**61.50 (50–71)**	**49.5 (44–55)**	** a, ** b, ** c, ** d, e **
Abstract thinking	2.00 (0–4.5)	4.00 (1–5)	**4.50 (0–5)**	2.00 (0–4)	**1.00 (0–4)**	0.00 (0–1)	* b
BT-ID							
Orientation	91.00 (48–109)	**103.00 (97–113)**	**109.00 (102–114)**	**91.00 (73–108)**	**46.00 (35–49)**	**28.50 (21–40)**	** a, ** b, ** c, ** d, ** e, ** f
Delay stories	2.00 (0–4)	3.00 (2–6)	**3.00 (2–5)**	3.00 (1–4)	1.50 (0–4)	**0.00 (0–2)**	* d, * e
Semantic fluency (eat/drink)	9.00 (6.5–12)	10.00 (9–12)	**10.5 (8–13)**	10.00 (8–11)	9.00 (5–13)	**6.00 (5–8)**	* d
Formal fluency	2.00 (0–4)	**3.00 (2–4**)	4.00 (2–6)	2.00 (1–4)	**0.00 (0–3)**	**0.00 (0–1)**	** b, ** c, *** d
Visual discrimination	18.00 (16–19)	18.00 (16–19)	18.00 (17–20)	18.00 (16–19)	16.00 (16–19)	18.00 (16–19)	

Values are given as median and range: GDS-DS, global deterioration scale for people with Down’s syndrome; BRIEF-P, behavior rating inventory of executive function parents’ form; BRI, behavioral regulation index; CAMCOG-DS, Cambridge cognitive examination for older adults with Down’s syndrome and other intellectual disabilities; BT-ID, Barcelona test for intellectual disability. * *p* < 0.05, ** *p* < 0.01, *** *p* < 0.001, according to Kruskal–Wallis test with Bonferroni correction. a, between cognitive/behavioral stability and mild Alzheimer’s disease; b, between cognitive/behavioral stability and moderate Alzheimer’s disease; c, between subjective cognitive/behavioral impairment and mild Alzheimer’s disease; d, between subjective cognitive/behavioral impairment and moderate Alzheimer’s disease; e, between mild cognitive/behavioral impairment and moderate Alzheimer’s disease; f, between mild cognitive/behavioral impairment and mild Alzheimer’s disease. Significant differences in bold.

**Table 8 ijerph-20-05096-t008:** Effect sizes between the GDS-DS stages.

	GDS-DS Stages							
	1–2		2–3		3–4		4–5	
	P	PS_est_	P	PS_est_	P	PS_est_	P	PS_est_
BRIEF-P								
BRI index	0.395	**0.60 ^a^**	0.061	**0.67 ^b^**	0.183	0.61	0.567	0.51
CAMCOG-DS								
Total score	0.938	0.51	**0.010 ***	**0.80 ^c^**	**0.027 ***	**0.69 ^b^**	0.088	**0.68 ^b^**
Abstract thinking	0.595	0.56	0.139	**0.63 ^a^**	0.549	0.51	0.129	0.65
BT-ID								
Orientation	0.314	0.62	**0.002 ****	**0.78 ^c^**	**0.000 *****	**0.94 ^c^**	0.057	**0.71 ^c^**
Free delay memory (stories)	0.775	0.53	0.663	0.54	0.331	0.55	0.467	**0.56 ^a^**
Semantic fluency (eat/drink)	0.856	0.52	0.265	0.60	0.906	0.49	0.114	**0.63 ^a^**
Formal fluency	0.364	0.61	**0.011 ***	**0.73 ^c^**	0.118	**0.66 ^b^**	0.716	0.44
Visual discrimination	0.733	0.54	0.367	0.58	0.363	**0.56 ^a^**	0.305	0.58

GDS-DS, global deterioration scale for people with Down’s syndrome; BRIEF-P, behavior rating inventory of executive function parents’ form; BRI, behavioral regulation index; CAMCOG-DS, Cambridge cognitive examination for older adults with Down’s syndrome and other intellectual disabilities; BT-ID, Barcelona test for intellectual disability. P, *p* value; PS_est_, probability of superiority effect size; * *p* < 0.05, ** *p* < 0.01, *** *<* 0.001; ^a^ PS_est_ ≥ 56, ^b^ PS_est_ ≥ 0.64, ^c^ PS_est_ ≥ 0.71. Significant effect sizes and *p* values with progressive decrease on the performance across the stages (Table 7) in bold.

## Data Availability

The data presented in this study are available on request from the corresponding author.

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
