# Peer review of "The Global Deterioration Scale for Down Syndrome Population (GDS-DS): A Rating Scale to Assess the Progression of Alzheimer’s Disease"

_ijerph, 2023, doi:10.3390/ijerph20065096_

Round 1

Reviewer 1 Report

This study examines the feasibility of Barry Reisberg’s Global Deterioration Scale adapted to adults with DS (GDS-DS) in the continuum of DS-AD (from stability to the mild-moderate-severe AD leaving out the very severe stage7). The novelty of the study lies in the fact that the GDS-DS also contemplates informant-rated subjective, behavioral and cognitive  changes. The findings are of extreme interest, but I do also have three serious concerns in order of importance:

1.     P. 11-12 Table 7 presents Cohen's d effect size estimates requires normally distributed data which is not the case in this study. The authors need to calculate effect size estimates with an alternative nonparametric statistic before stating in the Discussion on p. 12, lines 408-412 that a 1 SD decline on CAMCOG-DS and a 1 or 2 SD decline on the BT-ID Orientation subtest are of clinical relevance in a longitudinal follow-up.

2.     P. 12, lines 412-413:  The authors state that the purpose of the study was not to establish cut-off points for each GDS-DS stage. However, two out of the three mandatory criteria (BRIEF-P BRI; CAMCOG-DS) are based on cut-off points in function of the two ID levels.

3.     P. 6, lines 238-249 and Table 1, p. 5) In addition, the authors should explain the reason why the choice of the CAMCOG-DS total cut-off points differs for mild and moderate ID levels. For stage 3 CAMCOG-DS total cut-offs (mild ID) range from 82 to 69 based on their 2020  and 2013 studies for the upper and lower limits, respectively. By contrast, for moderate ID, CAMCOG-DS total cut-offs are derived from their 2020 study (i.e., ≤64-57 instead of ≤64-53). The same holds for the AD stages 4,5 and 6.  Why did the authors not report on basis of the neuropsychological tests and caregivers’ interviews a clinical profile (cognitive-behaviorial-functional) - separately for the two ID levels – characteristic for each stage like in the original GDS?

4.     P. 3, lines 134-138:  Subjective Cognitive Decline in the general population can result from many factors besides ageing, including anxiety, depression, sleep disorders, and drug side-effects (Balash Y, et al. Acta Neurol Scand. 2013; 127: 344–50). Similar confounding factors are also frequently seen in ageing DS and therefore individuals with the above characteristics have been excluded. The authors should explain in more detail the instruments they used to diagnose the presence and the severity of anxiety and depression and the criteria they applied to define “relevant drug treatment” (number of drugs?, anticholinergic and/or sedative burden?). On the other hand, having excluded mental health problems such as anxiety and depression the authors certainly have missed a number of individuals who could have entered in stage 2 and 3 (see NPI-DS by Dekker AD, et al. J Alzheimers Dis. 2021;81(4):1505-27). They also excluded individuals with hypothyroidism, considered a syndrome-specific disorder. The authors should provide more information (those without treatment because only recently discovered and not yet treated?, chronic hypothyroidism in replacement therapy?). The same holds true for VitB12 deficiency and anaemia.

Some minor observations

1.     P. 4, lines 161-162  The Tower of London listed among the objective cognitive evaluations was no longer mentioned in the results

2.     P. 9, lines 342-345. Only 27 out of 48 stable individuals in the PD group remained in the GDS-DS stages 1 and 2, whereas a small PD sample with AD diagnosis (n=11) increased to 31 individuals in the GDS-DS. The question arises which diagnostic criteria were used for stable, MCI and AD in the PD group? ICD-10, DC-LD, DSM-V, IASSID criteria? 

3.     P. 12, line 404: As regards the agreement between the two raters’ staging on the GDS-DS with the diagnosis in the PD group, kappa values ranged from substantial to excellent citing only one kappa value of 0.82 (line 403-404). What are the other “substantial” kappa values? This analysis really make sense since in the case of disagreement the two raters were allowed to discuss the case until agreement was obtained (p. 7, lines 301-302)?

4.     p. 12, line 435:  […] leading to a reduction in life expectancy. Evidence that ChEi’s positively interfere on survival is scanty and limited both in the general population (Balázs N, et al. Geroscience. 2022 Feb;44(1):253-63) and in people with DS-AD (Eady N, et al., Br J Psychiatry. 2018;212: 155-160.). A reduced life expectancy in people with DS due to AD (Iulita MF, et al., JAMA Netw Open. 2022 May 2;5(5):e2212910) is not a valid reason for the exclusion of  the GDS stage 7 (very severe AD) from the GDS-DS. Many people who develop DS-AD in their fifties without significant organic comorbidities survive up into the neurological phase of the disease (loss of speech, of the ability to walk and to sit without supports up to become completely bedridden)

GDS-DS has the potential to become an excellent staging instrument of DS-AD useful both in clinical (future) drug trials and nonpharmacological interventions and in daily clinical practice. However, the paper as actually presented needs major revisions to be considered for publication in Int J Environ Res Public Health.   

Reviewer 2 Report

Abstract- too many keywords. Kindly add some relevant keywords.

The research objective is clearly stated, and a good rationale for conducting the study. However, there are some areas of improvement needed-

In the methods section, clearly define the study population. Present the inclusion and exclusion criteria in a tabular format.

The final sample group information can be part of the results.

Where is the protocol for clinical assessment? This section is very vague. Needed further clarification.

Kindly add some information on ethical considerations.

Why Kruskal Wallis test was performed?

In the discussion, add some strengths of the research.

Round 2

Reviewer 1 Report

In this revised manuscript, the authors have well-addressed the concerns I raised except for the computation of the effect sizes estimates (GDS1-2, GDS2-3, etc.) [1] and the agreement between the two GDS-DS raters (JEC and JGA) [2].

[1] P. 9, lines 376-379. It is not clear which formula was used for the computation of the probability of superiority estimates with three levels of size (small-medium-large) quoted by Grissom (1993). In addition, the effect sizes displayed in Table 8 (P. 13) and in Appendix A are all inferior to the minimum requested value of 0.56 but one (Formal fluency GDS 4-5 with Mann-Whitney U test not statistically significant). I recommend the authors to read the attached.pdf file.  

[2] P. 8, line 348  the authors should clarify what they exactly mean with “[…] the case was discussed until  consensus was reached”. Was the inter-rater reliability test applied to the two raters’ first GDS-DS classification with a subsequent discussion of those cases with discordant decisions? If so, the inter-rater reliability test makes sense, otherwise the results are misleading. Of note, all reported Cohen’s weighted kappa values are greater than 0.80 indicating an almost perfect agreement, therefore delete “substantial” on line 449.

Some minor observations referring to both the original and revised manuscripts:

1.      P.2, line 86 the modified cued recall test was administered to preclinical, prodromal and mild AD-DS, therefore I suggest to replace “advanced” with full-blown (i.e., clinically manifest dementia albeit of mild degree)

2.      P.3, line 110 abbreviation of Barry Reisberg’s  Functional Assessment Staging is FAST

3.      P.3, line 125  behavioral instruments that are well established

4.      P.3, line 136 add alterations 

5.      P.3, line 143 replace “as” with represents or is  and “impairment” with decline from a previous level of efficiency

6.      P4, lines 157-158  I suggest to reword the sentence as follows: Finally, participants exposed to high anticholinergic burden through polypharmacy from psychotropics, gastrointestinal and cardiovascular medications were excluded […]

7.      P.5, line 181 not ICE-10 but ICD-10

8.      P.5, line 235 I believe “and/or loss of functionality” is more correct, see Table below.

9.     P.6, line 248 replace “compromise” with impairment or loss

10.  P.7, line 296 P.6, replace “affectation” with disorder or impairment or decline

11.  P.13, line 460 better Alzheimer’s dementia

12.  P.14, line 494 such as the cognitive […]

13.  P.14, line 502 Although not pathognomonic of AD, […]

14.  P.14, lines 507-508 “people with DS without impairment but with MCI and AD” is confusing, please reword

15.  P.15, lines 547-551  delete paragraph because it is a repetition of the previous one

16.  P.15, lines 552-554 some doubts about this affirmation; families are really able to understand the significance of the different GDS-DS stages exclusively based on outcomes of neuropsychological and other psychometric evaluations? Maybe in the future when enough data will be collected to construct specific clinical profiles in combination with functional assessment staging.

17.  P.16, line 574 not GDS-DS but GDS

18.  P.39, line 706  incomplete title  […] Primary Degenerative Dementia

Reviewer 2 Report

The manuscript can be accepted at this stage.

Author Response

Thank you very much for your kind help.

Round 3

Reviewer 1 Report

My compliments...but as stated in the discussion these data are  only a starting point for further and longstanding research!

See p.12, line 442 not "don't decreased"  but "did not decrease"; line 447 not "don't showed" but "did not show"